# Enrichment, Characterization, and Proteomic Profiling of Small Extracellular Vesicles Derived from Human Limbal Mesenchymal Stromal Cells and Melanocytes

**DOI:** 10.3390/cells13070623

**Published:** 2024-04-04

**Authors:** Sebastian Kistenmacher, Melanie Schwämmle, Gottfried Martin, Eva Ulrich, Stefan Tholen, Oliver Schilling, Andreas Gießl, Ursula Schlötzer-Schrehardt, Felicitas Bucher, Günther Schlunck, Irina Nazarenko, Thomas Reinhard, Naresh Polisetti

**Affiliations:** 1Eye Center, Medical Center, Faculty of Medicine, University of Freiburg, Killianstrasse 5, 79106 Freiburg, Germany; 2Faculty of Biology, University of Freiburg, Schaenzlestrasse 1, D–79104 Freiburg, Germany; 3Institute of Surgical Pathology, Faculty of Medicine, Freiburg, Medical Center, University of Freiburg, 79085 Freiburg im Breisgau, Germany; 4Department of Ophthalmology, University Hospital Erlangen, Friedrich-Alexander-University of Erlan-gen-Nürnberg, Schwabachanlage 6, 91054 Erlangen, Germany; 5Institute for Infection Prevention and Hospital Epidemiology, Faculty of Medicine, Medical Center, University of Freiburg, 79106 Freiburg, Germany

**Keywords:** limbal melanocytes, limbal mesenchymal stromal cells, extracellular vesicles, exosomes, limbal epithelial progenitor cells, limbal stem cell niche, proteomics

## Abstract

Limbal epithelial progenitor cells (LEPC) rely on their niche environment for proper functionality and self-renewal. While extracellular vesicles (EV), specifically small EVs (sEV), have been proposed to support LEPC homeostasis, data on sEV derived from limbal niche cells like limbal mesenchymal stromal cells (LMSC) remain limited, and there are no studies on sEVs from limbal melanocytes (LM). In this study, we isolated sEV from conditioned media of LMSC and LM using a combination of tangential flow filtration and size exclusion chromatography and characterized them by nanoparticle tracking analysis, transmission electron microscopy, Western blot, multiplex bead arrays, and quantitative mass spectrometry. The internalization of sEV by LEPC was studied using flow cytometry and confocal microscopy. The isolated sEVs exhibited typical EV characteristics, including cell-specific markers such as CD90 for LMSC-sEV and Melan-A for LM-sEV. Bioinformatics analysis of the proteomic data suggested a significant role of sEVs in extracellular matrix deposition, with LMSC-derived sEV containing proteins involved in collagen remodeling and cell matrix adhesion, whereas LM-sEV proteins were implicated in other cellular bioprocesses such as cellular pigmentation and development. Moreover, fluorescently labeled LMSC-sEV and LM-sEV were taken up by LEPC and localized to their perinuclear compartment. These findings provide valuable insights into the complex role of sEV from niche cells in regulating the human limbal stem cell niche.

## 1. Introduction

Corneal epithelial homeostasis and regeneration are crucial for the preservation of corneal transparency and visual acuity. These processes rely on the presence and activity of limbal epithelial progenitor cells (LEPC) situated within the basal epithelial layers of the corneoscleral limbus [1]. Proper LEPC functionality and self-renewal capacity are dependent on a particular niche milieu, which is characterized by specific physical features, autocrine and paracrine signals, the presence of several niche cell types, including limbal mesenchymal stromal cells (LMSC) and limbal melanocytes (LM), as well as a distinct extracellular matrix (ECM) composition [2,3,4,5,6]. Recent reports showed that co-cultures of LEPC with LMSC or LM supported proliferation and migration and reduced further differentiation of LEPC [5,7,8,9]. Moreover, it has also been reported that the LM and LMSC secretome supports corneal epithelial regeneration, and modulates angiogenic and inflammatory responses [8,10]. This indicates that along with direct cell–cell interactions, indirect paracrine mechanisms involving soluble growth and signaling factors also play a crucial role in maintaining LEPC niche homeostasis. In animal models of limbal stem cell deficiency, LMSC-conditioned media (CM) induced substantial re-growth of corneal epithelial cells on the corneal surface with fewer cells attaining a conjunctival phenotype [11,12]. This has led to the concept of using the LMSC secretome, a mixture of extracellular vesicles (EVs) and paracrine soluble factors, as an alternative to LMSC therapy in regenerative medicine [12]. Furthermore, there is increasing evidence that EV transport proteins, lipids, and nucleic acids to recipient cells via paracrine and autocrine mechanisms [13,14,15], which can reprogram target cells and modulate the phenotype of recipient cells [16].

EVs are lipid bilayer-bound vesicles secreted by all cell types into extracellular space. They are heterogeneous in composition and function [17,18]. In general, EVs can be categorized as small EVs (sEVs) (40–150 nm), microvesicles (100–1000 nm), and apoptotic bodies (1000–5000 nm) based on factors such as size, expression of transmembrane markers and biogenesis pathways. sEVs, derived from endosomes, have gained greater scientific research attention as they act as a functional mediator of intercellular communication associated with both physiological and pathological processes [19,20]. Moreover, these vesicles are attracting significant clinical interest as they offer the potential for disease diagnosis via liquid biopsy and therapeutic interventions [21]. The isolation and characterization of sEVs released by limbal niche cells, namely LMSC and LM, is only in its beginning. Recent studies indicated that LMSC-derived sEVs promoted proliferation and migration of LEPC in vitro and improved epithelial wound closure in a mouse model [22,23,24,25]. Moreover, LMSC-sEVs also exhibited anti-inflammatory, immunomodulatory, and anti-angiogenic properties, both in vitro and in vivo [24,25,26]. However, while LMSC-sEV have been studied to some extent, data on LM-derived sEVs are completely lacking.

In order to shed light on this underexplored area, sEVs were isolated from the CM of LMSC and LM using tangential flow filtration (TFF) and size exclusion chromatography (SEC). The isolated sEV populations were characterized by nanoparticle tracking analysis (NTA), Western blot (WB), and electron microscopy. In addition, their protein cargo was determined by mass spectrometry and Multiplex bead arrays.

## 2. Results and Discussion

### 2.1. Enrichment and Characterization of sEVs

#### 2.1.1. Cell Isolation

Limbal cluster-derived cell suspensions from human donor corneas provided a yield of 1.4 ± 0.4% of CD117^+^ cells (LM) and 1.9 ± 0.7% of CD90^+^ cells (LMSC) and the remaining CD117^−^CD90^−^ cells (LEPC); gates were set based on isotype controls (Appendix A). Primary cell cultures (Appendix A) presented cell type-specific morphological phenotypes as published earlier [9,27].

#### 2.1.2. sEV Enrichment

sEVs were produced in vitro by LMSC or LM under defined cell culture conditions. LMSC-sEVs were produced using an animal-component-free culture medium. In contrast, LM-sEVs were generated in CNT-40 medium containing 1% fetal bovine serum (FBS) supplement. Cell viability was assessed before and after 72 h of exposure to the EV-harvesting media using trypan blue exclusion staining, revealing a robust viability of 96.4 ± 2.3%. Given the well-recognized impact of purification methods on sEV quality [28], we employed a purification approach combining TFF and SEC [29] to enrich sEVs from CM of both cell populations (Figure 1A). Initial centrifugation at 2500× *g*, followed by 0.8 µm and 0.2 µm filtration of CM facilitated depletion of apoptotic bodies and large particles (>200 nm) (Figure 1A). Subsequent TFF concentration and SEC fractionation facilitated the isolation of highly enriched sEV populations with minimal co-isolated non-EV membrane vesicles [29].

#### 2.1.3. Biophysical Properties

According to the guidelines suggested by the International Society of EV (MISEV2018), we determined essential biophysical and biochemical properties of the sEV fractions [30], namely, particle size and concentration by NTA, protein concentration by micro-BCA assay, and morphology by transmission electron microscopy (TEM), and EV marker proteins were detected by WB and flow cytometry, and total protein cargo patterns were profiled by mass spectrometry. NTA showed the highest number of particles in fraction 3 (95.8 × 10^9^ (±17.8) particles/mL for LM-sEVs and 67.0 × 10^9^ (±13.0) particles/mL for LM-sEVs) followed by fraction 4 (34.2 × 10^9^ (±13.7) and 33.4 × 10^9^ (±9.2) particles/mL, respectively) and fraction 2 (27.3 × 10^9^ (±5.3) vs. 16.4 × 10^9^ (±5.0) particles/mL, respectively) (Figure 1B). Particle number was significantly higher in the CM compared to the unconditioned medium for both cell types, especially in fraction 3 (F3) (Figure 1B). The absence of particles in the flow through (FT) or void volume (V) following sample preparation by TFF and SEC, demonstrates complete particle retention throughout these processes. Protein concentration analysis by micro-BCA revealed the highest protein concentrations in F7, F8, and flow through (FT) in both conditioned and unconditioned (control) media (Figure 1C). A higher particle-to-protein ratio indicates lower contamination in EV preparation [30,31]. F3 of both LMSC-(3.4 × 10^9^ (±1.6) and LM-sEVs (7.1 × 10^8^ (±2.0)) had the highest ratios compared to other fractions (Figure 1D), and the purity was in line with previous studies [31,32]. NTA analysis revealed that LMSC-sEVs had an average peak size at 133.9 ”m (±1.2), whereas LM-sEVs averaged at 112.8 nm (±19.2) (Figure 1E), similar to previous reports on corneo limbal keratocytes and human epidermal melanocytes [22,33]. The average peak size of other fractions has provided in Appendix A.

#### 2.1.4. Western Blot Analysis

WB analysis was performed to characterize the different fractions by examining the presence of transmembrane proteins (CD9, CD63, CD81), the cytosolic EV protein Alix (PDCDIP), cell type-specific markers (CD90 for LMSC and Melan-A for LM), and the non-EV marker calnexin (CANX) as well as the co-isolated protein bovine serum albumin (BSA) [30]. WB analysis revealed that Alix was present in the majority of the fractions (at least as a faint band) but with a stronger band in F3 (Figure 2A). Similarly, CD9, CD63, and CD81 were present in fractions 2–4 but predominantly found in F3 (Figure 2A). The LMSC cell marker CD90 was present in fractions 2–4 of LMSC-sEV with the strongest signal in F3 (Figure 2B). Likewise, the LM-specific Melan-A appeared as a robust band in F3 of LM-sEVs (Figure 2B). Unconditioned media did not show any evidence of either EV- or cell-specific markers (Appendix A). The absence of the endoplasmic reticulum-resident protein CANX across all LMSC- and LM-sEV fractions (Figure 2C) but its presence in the cell lysates (Appendix A), validated the effective isolation of sEVs without contamination by other cellular components [30]. WB analysis detected residual BSA, an abundant fetal bovine serum constituent, across LM-sEV fractions (Figure 2C) as well as in the unconditioned media control (Figure 2D), but not in the LMSC-sEV (Figure 2C,D). Since BSA signals were stronger in later eluting LM-sEV fractions 7 and 8 compared to earlier fractions 2–4, eliminating serum or switching to EV-depleted formulations could further enhance purity [34]. While standardized CNT-40 enables robust primary LM propagation in vitro [7,9,27,35], the effects of shifting this medium to a serum-free formulation on maintaining LM phenotype would need to be studied before follow-up studies on LM-sEV. Since these data confirmed enrichment of sEV in F3, this fraction was used in subsequent experiments.

#### 2.1.5. Electron Microscopy

TEM analysis revealed that particles present in F3 exhibited a classical EV morphology with an intact membrane and a well-described doughnut or cup shape (Figure 2E) and sizes of 55.4 ± 17.0 nm for LM-sEVs and 68.0 ± 23.7 nm for LMSC-sEVs (Figure 2F). Substantially, larger EV diameters were determined with the NTA device as compared to TEM-based size measurements, which is consistent with earlier studies [36,37]. This discrepancy is likely due to shrinkage effects of formaldehyde fixation during TEM sample preparation, which was reported to lead to an underestimation of the true EV size by up to 21%.

### 2.2. Flow Cytometry Bead Assays

To further characterize sEV surface epitopes, we performed a multiplex bead-based flow cytometry assay on sEV samples (F3). The assay comprises 39 hard-dyed capture bead populations (4 µm diameter) each of the beads coated with monoclonal antibodies against one of 37 potential EV surface antigens or one of two internal isotype negative controls (Appendix A). The markers were considered positive/detected if their nMFI was greater than 2% in all samples analyzed (n = 6).

In accordance with our Western blot results, the commonly used EV markers CD9 (LMSC—77.5 ± 5.8; LM—95.1 ± 3.0%), CD63 and CD81 (>95.0% in both LMSC- and LM-sEVs) were highly expressed (Figure 3A). Other commonly detected markers in LMSC- and LM-sEVs include CD29 (LMSC—37.2 ± 2.4%; LM—28.7 ± 5.6%), CD44 (LMSC—42.4 ± 3.2%; LM—61.8 ± 4.5%), CD49e (LMSC—32.4 ± 4.3%; LM—15.0 ± 3.3%), CD105 (LMSC—3.7 ± 0.3%; LM—13.8 ± 1.6%) and CD146/melanoma-cell adhesion molecule (MCAM, LMSC—4.4 ± 1.8%; LM—92.4 ± 11.9%), but the levels varied between cell types (Figure 3A). Markers exclusively detected on LM-sEVs included CD41b (15.9 ± 5.7%), HLA-ABC (5.3 ± 1.4%), melanoma-associated chondroitin sulfate proteoglycan (MCSP) (77.6 ± 3.7%) and ROR1 (3.6 ± 0.8%) (Figure 3A). To confirm differential detection, CD146 and MCSP were further analyzed by WB, validating the predominant levels of both markers in LM-sEVs compared to LMSC-sEVs (Figure 3B). It has been reported that CD146 is expressed in over 70% of metastatic melanoma cells, while it is notably absent in normal melanocytes in vivo [38]. To further address the observation of CD146 in sEVs, immunohistochemical analysis of organ-cultured limbal tissue sections, alongside WB and immunostaining of samples from cultured cells was performed. The analysis revealed the absence of CD146 staining (green) in melanocytes (red, gp100^+^) and stromal cells (cyan, vimentin^+^) in fresh tissue (Figure 3C), whereas cultured LM and LMSC demonstrated robust CD146 expression by immunoblotting (Figure 3B) and immunocytochemistry (Appendix A). These data suggest that CD146 protein levels are upregulated in cell culture, which aligns well with the observation that endothelin-1 stimulates CD146 protein expression in primary human melanocytes [38]. Given that the proprietary CNT-40 medium used for LM expansion contains endothelin-1 as a supplement [7], this likely explains the marked detection of CD146 in LM-sEVs. Interestingly, while ex vivo cultured LMSC exhibited CD146 staining, CD146 was conspicuously minimal to absent (variation in biological samples) in LMSC-sEV isolates (Appendix A). This finding warrants further mechanistic investigation in future studies. Similarly, normal melanocytes express little or no MCSP in the limbal niche as confirmed in our immunohistochemical stainings (Figure 3C), whereas increased levels of MCSP were detected in melanoma cells as well as in melanoma EVs [39]. Interestingly, cultured LMSC and LM were found to express MCSP (Figure 3B, Appendix A), whereas its detection in sEVs was restricted to LM-sEV and non-vesicular fractions (low vesicle content fractions) F5 to F8 in LMSC-sEVs (Appendix A), suggesting that its secretion occurs independently of Evs in LMSC cultures. MCSP is a well-established melanoma immunotherapy target, and was shown to promote corneal vascularization in vivo, suggesting a role in angiogenesis [40]. These results indicate that upregulated expression of metastatic melanoma markers CD146 and MCSP in cultured LM likely stems from induction by the culture conditions and the inherent plasticity of cultured melanocytes, rather than malignant transformation [7].

### 2.3. Proteomics Analysis

To obtain an unbiased inventory of sEV proteins, quantitative proteomic profiling was performed. Quantitative mass spectrometry proteomic profiling identified 511 and 466 unique proteins based on LFQ intensity in sEVs from LM and LMSC, respectively (n = 5 for each). Principal component analysis (PCA) revealed a clear separation between LM and LMSC-sEV samples (Appendix A), and unsupervised hierarchical clustering, as depicted in the heatmaps (Appendix A), further highlighted the distinct clustering of samples. Further analysis focused on proteins detected in ≥3 biological replicates per group (Appendix A (LM-sEVs) and Appendix A (LMSC-sEVs)). The analysis was initially focused on the general sEV proteomes LMSC and LM, followed by a comparative analysis to identify differentially enriched proteins exclusive to LMSC-sEVs or LM-sEVs.

#### 2.3.1. General sEV Proteomes

The datasets revealed proteins (identified by official gene symbols) associated with the plasma membrane (PM) and/or endosomes in both LMSC- and LM-sEVs. These included tetraspanins (CD63, CD81, CD82, CD9), 5′ nucleotidase (NTE5), integrins (ITG) (αv, β1), multipass membrane proteins (BSG, ADAM10), a complement binding protein (CD59) (Figure 4A(i)); cytosolic proteins including endosomal sorting complexes required for transport (ESCRT)-1 complex protein (TSG101), ALIX (PDCD6IP), annexins (A2, A6), syntenin (SDCBP), and heat shock protein (HSP90AB1, HSPA8) (Figure 4A(ii)), which are involved in EV formation, recipient cell binding, and membrane fusion [30]. However, intensity patterns of these proteins varied between LMSC- and LM-sEVs (Figure 4A(i,ii)). Proteomics data aligned with WB in detecting the EV markers CD63, CD9, CD81, and Alix (Figure 2A) and with the flow cytometric bead array (Figure 3A) in detecting CD63, CD9, CD81, and ITG β1 (CD29). Notably, ITG β3 (ITGB3) was exclusively found in LM-sEVs. It contributes to the formation of integrin αvβ3 (vitronectin receptor, also binding to fibronectin) expressed in melanocytes and melanoma [41]. The ITG α2 (ITGA2), α3 (ITGA3), α5 (ITGA5), were uniquely found in LMSC-sEVs (Figure 4A(i)). While CD41b (ITGA2b) and ITGA5 were detected in the flow cytometry bead assay, they were not identified in our proteomic profiling of LM-sEVs. This discrepancy might arise from differences in the sensitivity and specificity of the methodologies employed or potential non-specific antibody binding in the bead arrays. However, we agree that additional investigations are necessary to validate and elucidate the presence or absence of CD41b and ITGA5 in LM-sEVs conclusively. Furthermore, EV encapsulated known lineage-specific antigens [9], such as CD90 (THY1) in LMSC-Evs and melanocyte proteins melan-A (MLANA), tyrosinase-related protein 1 (TYRP1/TRP1), and premelanosome protein (PMEL) in LM-sEVs (Figure 4A(i)), indicating enrichment of distinct vesicle signatures generated by LMSCs and LMs. The detection of CD90 in LMSC-sEVs and MLANA and TYRP1 in LM-sEVs aligned with WB data (Figure 4B). Along with previous studies [42,43], actins (ACTB, ACTG1), tubulin (TUBB), and the enzyme (GAPDH) were identified in both LMSC- and LM-sEV (Figure 4A(ii)). Further validation by Western blot revealed presence of β-actin (ACTB) both in LM-sEVs and LMSC-sEVs (Figure 4B). These data suggest that while common vesicle-associated proteins were present, specialized integrins, receptors, and lineage antigens unique to LMSC or LM are incorporated in sEVs, indicating discrete sEV cargo composition aligned with differential functional activities of stromal and pigmented niche cells.

##### Assessment of sEV Purity

To evaluate sEV purity, quantification of common contaminants like apolipoprotein is required [30]. Apolipoproteins A1/2 (APOA1/2) and B (APOB) were absent in LMSC-sEVs, and only minor APOB was detected in LM-sEVs (Figure 4A(iii)), arguing for distinct lipoprotein profiles [44]. Minimal amounts of APOB may stem from LM culture medium containing 1% FBS [45,46,47]. However, adding additional purification steps might reduce possible APOB contamination but risks reducing EV yields [48]. Moreover, epidermal melanocytes have the capacity of cholesterol signaling via the APOB100/LDL receptor, which promotes melanogenesis in vitro [49], suggesting that the detected APOB may represent either physiological pathway components or vesicles supporting pigment production rather than mere cell culture medium-derived contaminants. Importantly, LMSC- and LM-sEVs lacked markers related to the endoplasmic reticulum (CANX), Golgi apparatus (GOLGA2), mitochondria (CYC1), or autophagosomes (ATG9A), as well as cytoskeletal KRT18 (Figure 4A(iv)), distinguishing them from debris such as apoptotic bodies or shed membrane components [30,50]. The lack of CANX in LMSC- and LM-sEVs is supported by WB data (Figure 2). Interestingly, nuclear histones (H2A (HIST1H2AC), H3 (HIST1H3A), H4 (HIST1H4A)) were exclusively detected in LM-sEVs (Figure 4A(iv)). Although histone enrichment in LM-sEVs could indicate possible contamination, given reports of histones as abundant sEV proteins [51,52,53] and their cytoplasmic trafficking (e.g., H4) [54], the identification of histones is in line with the possibility of inclusion of nuclear components in Evs of endosomal origin [30].

##### Functional Enrichment Analysis

Gene ontology (GO) and Kyoto Encyclopedia of Genes and Genomes (KEGG) pathway enrichment analysis of the sEV proteomic data sets was implemented in R using respective software packages to elucidate associated biological processes (BP), molecular functions (MF), cellular components (CC), and signaling pathways (Appendix A (LM-sEVs) and Appendix A (LMSC-sEVs). The top 10 enriched BP, CC, MF, and KEGG pathways based on significance are shown in Figure 4C,D. A proteomic analysis revealed that LM-sEVs and LMSC-sEVs are enriched in proteins involved in focal adhesion, extracellular matrix interactions, wound healing, and cytoskeletal regulation (Figure 4C). Pathway analysis highlighted the roles of LM-sEVs in focal adhesion and cell–ECM interactions and of LMSC-sEV in focal adhesion, ECM interactions, and PI3K-Akt signaling (Figure 4D). These data suggest that both LMSC- and LM-sEVs play distinctive roles in cell–matrix interactions, cell adhesion signaling, and cytoskeletal regulation. This may highlights their crucial involvement in intercellular communication and sensing of the microenvironment. It is noteworthy that cells have the capacity to influence the behavior of neighboring cells by providing them with altered ECM [55,56,57].

#### 2.3.2. Comparative sEV Proteome Profiling

The Venn diagram demonstrates that 158 proteins were found in both LMSC- and LM-sEVs, whereas 177 proteins were exclusively present in LM-sEVs and 143 proteins in LMSC-sEVs, respectively (Figure 4E). Many of the proteins found exclusively in LM-sEVs are related to pigmentation and cell structures, such as melanosome pigment granules, focal adhesion and cell–substrate junction (Figure 5A, Appendix A) involved in cellular pigmentation, development, and vesicle organization (Figure 5A), in keeping with the specialized function of the parent melanocyte [58]. KEGG pathway analysis highlighted connections of LM-sEV to motor proteins, phagosome, and pathogenic Salmonella and Escherichia coli (Figure 5A, Appendix A). On the other hand, proteins found exclusively in LMSC-sEV are predominantly related to the ECM and cell adhesions, including focal adhesions. They play roles in collagen biosynthesis, cell-matrix adhesion, and collagen fibril organization (Figure 5A, Appendix A), reflecting LMSC specialization in ECM maintenance and cell–ECM communication in the limbal stem cell niche [4,6].

##### Melanosome-Related Proteins and Pigmentation

LM are considered to transfer melanin granules to neighboring LEPC, protecting them from UV damage. The pigmentation degree correlates with LEPC immaturity, with highly pigmented melanocytes associating with the most primitive states [59]. Along with classical melanosomal markers tyrosinase (TYR) (Figure 5B), TYRP1, and PMEL (Figure 4A(i)), LM-sEVs also contain RAB GTPases (RAB11A, RAB11B, RAB1A, RAB27A, RAB32, RAB38, RAB5C, RAB6A, RAB6B) (Figure 5B) involved in exosome formation by directing multivesicular bodies to the plasma membrane for secretion [60], while also regulating melanosome biogenesis and transport [61]. Additionally, the late endosomal/lysosomal protein LAMP1 [62] was detected (Figure 5B) in line with previous EV studies [63,64]. LAMP1 expression is stimulated during melanogenesis, with TRP1 localizing on melanosome membranes and conferring protection from toxic intermediates produced alongside active tyrosinase [65]. Proteomic profiling further revealed the presence of apolipoprotein-E (APOE, Figure 5B), known to be associated with late endosomes and vesicles, mediating amyloid fibril formation by the pigment cell-specific PMEL, and facilitating its sorting into intraluminal vesicles [66]. UV exposure can lead to corneal damage and ocular surface disorders by inducing oxidative stress, DNA damage, and inflammation in corneal cells, causing dysregulated cell death, proliferation, and differentiation [67,68]. These cellular aberrations contribute to the development of conditions like pterygium and ocular surface cancers [67,68]. Our findings suggest that LM-sEVs may protect against UV-induced damage by modulating these cellular processes through the transfer of protective biomolecules like antioxidants, anti-inflammatory factors, and regulatory RNAs. Elucidating the vesicular components could reveal the mechanisms underlying the potential protective role of LM-sEVs against UV-induced corneal damage.

##### Extracellular Matrix Proteins and Niche Regulation

The ECM architecture of the limbal niche, enriched in collagens, glycoproteins and proteoglycans, is critical for niche functions such as providing cell adhesion sites, imparting mechanical integrity, serving as an interstitial space, conveying signals, and enabling communication between cells [4,69,70]. Emerging studies reveal that Evs can be considered structural and functional ECM components which participate in matrix organization, regulation of embedded cells, and determination of the physical properties of various connective tissues [56,71]. In the present study, the main basement membrane (BM) collagens, collagen type (COL) XVIIIα1 (COL18A1) and COLIV α1/α2 (COL4A1/2), were exclusively present in LM-sEVs, whereas fibrillar collagens, COL IIIα1 (COL3A1) and COLV α1 (COL5A1), were found in LMSC-sEVs (Figure 5B). Alpha-2 chains of COL IV and COLXVI have been previously reported to preferentially localize to the limbal BM [4,70]. Moreover, COL IV promotes the ex vivo expansion of LEPC and in vitro differentiation of embryonic stem cells towards a corneal epithelial fate [72]. Multiple laminin (LN) subunits including α1 (LAMA1), β1 (LAMB1), β2 (LAMB2), and γ1 (LAMC1) chains were identified in sEVs of both cell types. Intriguingly, the limbus-specific LN-α5 (LAMA5) which forms LN10/11 and promotes the growth of LEPC and LM [7,73], was exclusive to LM-sEVs. The laminin α2 (LAMA2), α4 (LAMA4), and γ2 (LAMC2) chains were exclusively found in the LMSC-sEVs. Additional ECM glycoproteins present in both sEV populations included fibronectin (FN1), nidogen 1 (NID1), thrombospondin 1 (THBS1), transforming growth factor β-inducible (TGFBI), milk fat globule-EGF factor 8 (MFGE8), and galectin-3-binding protein (LGASL3BP), whereas the glycoproteins decorin (DCN), elastin microfibril interfacer 1 (EMILIN1), fibrillin 1 and 2, NID2, SPARC, and Tenascin-C (TNC) were exclusively found in the LMSC-sEVs. Of note, TGFBI was highly enriched in LMSC-sEVs vs. LM-sEVs, and mutations in TGFBI cause accumulation of cloudy material in the superficial cornea, leading to impaired vision and painful erosion due to poor epithelial adhesion [74,75]. The proteoglycan agrin (AGRN) was found in sEVs of both cell types, whereas versican (VCAN), aggrecan (ACAN), and biglycan (BGN) was found only in LMSC-sEVs, and brevican (BCAN) and BM-specific perlecan (HSPG2) were exclusively detected in LM-sEVs (Figure 5B). Thus, proteomic profiling reveals specialized ECM protein signatures in LM- and LMSC-sEVs consistent with the distinct niche microenvironments and progenitor cell regulation tasks of these cell types. The cell-surface proteoglycan CD44 was found in sEVs from both cell types, in line with flow cytometric bead arrays, whereas glypican 1 was present specifically in LMSC-sEVs (Appendix A), and the proteoglycans MSCP and glycoprotein CD146 (MCAM) were present only in LM-sEVs. The exclusive detection of MCSP and CD146 in LM-sEVs by mass spectrometry further validates the multiplex bead array (Figure 3A) and immunoblot data (Figure 3B) demonstrating enrichment of these markers specifically in vesicles of melanocytic origin.

##### Growth Factors, Cytokines, and Matrix Remodeling Proteins

Accumulating evidence indicates that growth factors and soluble mediators are associated with the EV surface, implicating Evs in systemic dissemination of bioactive growth factors [76]. Differential enrichment analysis revealed distinct mitogens, such as TGFB1, platelet-derived growth factor C (PDGFC), inhibin subunit β A (INHBA), and colony-stimulating factor 1 (CSF1) as exclusively present in LMSC-sEVs (Figure 5B), while growth/differentiation factor 15 (GDF15) was only present in LM-sEVs (Figure 5B). GDF15 expression increases in melanocytes following ultraviolet radiation, suggesting its involvement in pigmentation pathways [77]. Beyond growth factors, we identified matrix metalloproteinases (MMPs) and tissue inhibitors of metalloproteinases (TIMPs) as sEV cargo, which regulate ECM proteolytic remodeling. Notably, MMP2 was common in both sEV populations, whereas TIMP1, TIMP2, and TIMP3 were exclusive to LMSC-sEVs (Figure 5B), indicating their preferential involvement in limbal niche ECM turnover [56,78]. Our proteomic results also revealed that LMSC-sEVs carried proteins relevant to wound healing and angiogenesis including thrombospondins (THBS2) [79], latent-transforming growth factor beta proteins (LTBP1 and 2), angiopoietin-like 4 (ANGPTL4) [80], serine protease inhibitor clade E member 1 (SERPINE1), and A disintegrin and metalloproteinase domain-containing protein 12 (ADAM12) [81] (Figure 5B). TGF- β1 is known to trigger fibrosis and myofibroblast transdifferentiation and was found exclusively in LMSC-sEV along with TGF-β1 binding proteins LTBP1 and LTPBP2 (Figure 5B). Proteomic analysis revealed the presence of several proteins known to mediate epithelial migration including neuropilin-1 (NRP1), EPH receptor A2 (EPHA2), Ras-related protein (RRAS), secretogranin II (SCG2), and follistatin-related protein 1 (FSTL1) [82,83] (Figure 5B). Thus, proteomic profiling reveals that limbal niche-derived Evs carry selective repertoires of pro-angiogenic factors, matrix remodeling enzymes, and wound healing mediators, which implies coordinated functional roles of sEVs in corneal ECM homeostasis and vascularization.

Overall, our study provided insights into the roles of sEVs derived from different cellular sources within the limbal stem cell niche. We found that LMSC-sEVs contribute to ECM deposition, influencing integrin-dependent signaling and differentiation of neighboring cells. In contrast, LM-sEVs predominantly support melanosome distribution and oxidative stress mitigation. Further investigations into the molecular cargo and mechanisms of these sEVs, as well as their potential therapeutic applications, are warranted.

### 2.4. Uptake of Isolated sEVs by Recipient Cells

A crucial aspect of any protocol designed for the isolation of Evs is the preservation of their functional properties. We verified that our isolation protocol did not disrupt the ability of Evs to be incorporated into recipient cells. To this end, we stained isolated Evs (F3) with a lipophilic fluorophore (Memglow). Following 3 or 6 h of incubation, a flow cytometric analysis revealed that LEPC treated with CM-derived sEVs (LMSC-sEVs or LM-sEVs) showed increased fluorescence intensity compared to LEPC treated with unconditioned media-derived sEVs as well as untreated control LEPC (Figure 6A). Furthermore, confocal microscopy revealed the presence of LM- or LMSC-sEV label signals in the perinuclear region of LEPC after 6 h (green, Figure 6B), strongly suggesting the internalization of sEV by LEPC [22]. LEPC were co-stained for F-actin with Phalloidin-TRITC (red, Figure 6C) for further confirmation. These data substantiate the potential role of sEVs in mediating communication between LEPC and LM or LMSC, suggesting that sEVs participate in paracrine signaling within the limbal stem cell niche. However, additional functional assays are required to prove the concept of sEV involvement in limbal niche homeostasis.

## 3. Conclusions

This study successfully isolated sEV from two key cell types in the limbal stem cell niche, LMSC and LM, and showed their internalization in LEPC, indicating the preservation of sEV functionality. Proteomic analysis revealed that these vesicles may play signaling roles related to ECM regulation and intercellular communication within the niche microenvironment as they carry distinct subsets of ECM molecules and proteins involved in ECM modulation. In the niche environment, LMSC-sEV appear to contribute to ECM deposition, influencing cell signaling and differentiation, whereas LM-sEV predominantly support melanosome distribution and oxidative stress mitigation. Further analysis is needed to elaborate the specific functional relevance of the isolated sEVs in mediating limbal niche homeostasis.

## 4. Limitations of the Study

While this study provides valuable insights into the distinct sEV populations derived from LM and LMSC through comprehensive characterization techniques, certain limitations should be acknowledged. The reliance on in vitro cultured cells raises concerns about the extent to which the observed sEV profiles accurately reflect the in vivo limbal niche environment. Additionally, the use of organ-cultured corneas rather than fresh limbal tissues for cell isolation may have influenced the cellular properties and, consequently, the EV characteristics. The use of serum-containing media for LM culture may have introduced contaminants like BSA into the isolated LM-sEVs. Moreover, the study lacked functional assays to directly demonstrate the role of LM-sEVs and LMSC-sEVs in regulating limbal niche homeostasis or epithelial regeneration, limiting the functional implications that can be drawn. Addressing these limitations in future studies could help reinforce and expand the findings regarding the specialized roles of LM-sEVs and LMSC-sEVs in the limbal stem cell niche microenvironment.

## 5. Materials and Methods

Human donor corneoscleral tissues with appropriate research consent were obtained by the Lions Cornea Bank Baden-Württemberg after retrieving of corneal endothelial transplants [84], with informed consent from donors or relatives, and the experiments using human tissues were approved by the University of Freiburg’s Institutional Review Board of the Medical Faculty (25/20) in adherence to the Declaration of Helsinki.

### 5.1. Cell Isolation and Culturing

LEPC, LMSC, and LM were isolated from the limbus of organ-cultured corneoscleral tissue (*n* = 215, mean age 71.7 ± 12.7 years; culture duration 24.0 ± 5.8 days; post-mortem time 30.6 ± 19.5 h) as previously described [9].

### 5.2. Preparation of Conditioned Media and EV Isolation

LMSC (P1) and LM (P1) were seeded into T75 flasks (four flasks/preparation) and cultured up to 70–80% confluency in their respective media as mentioned above. Then, the cell monolayers were washed with DPBS twice and the Mesencult media was replaced with Mesencult ACF media for LMSC, while LM received the same CNT-40 media to generate conditioned media. Unconditioned media served as negative control. After 72 h, both conditioned and unconditioned media were collected and centrifuged at 2.500× *g* for 15 min at 4 °C to remove any cells or cell debris. Then, the supernatant was subjected to sequential filtration using an 0.8 µm filter followed by a 0.2 µm syringe filter to eliminate large Evs (>200 nm). Filtered media were concentrated up to 1 mL using TFF (TFF-Easy; Hansa Biomed, Tallin, Estonia) followed by 3× PBS wash. Concentrated media was then loaded twice (500 µL each time) into a SEC (qEVoriginal/35 nm Gen 2; IZON Science, Christchurch, New Zealand) using 0.2 µm-filtered DPBS (Invitrogen, Waltham, MA, USA) as the running buffer. First, 2.0 mL was discarded as void volume, and right after, 500 µL fractions (F1–F8) were collected and stored at −80 °C until further use.

### 5.3. Characterization of LMSC and LM sEVs

#### 5.3.1. Measurement of Particle Count and Total Protein Concentration

The size distribution curves and concentration of particles (particle count) present in LMSC and LM EV fractions (F1–F8) were determined by NTA using the ZetaView BASIC PMX-120 instrument (Particle Metrix GmbH, Inning am Ammersee, Germany) equipped with NTA 2.0 analysis software. Using 100 nm standard beads, the instrument was calibrated, and the following settings were used: positions—11; cycles—1; minimum size—5 nm; maximum size—150 nm; trace length—15 s; sensitivity—80%; shutter speed—75 ms; frame rate—30; and temperature 23–24 °C. Diluted LMSC- and LM-sEVs fractions (100- to 5000-fold) were loaded into the NTA instrument, and the values were recorded. Background measurements were performed with filtered PBS, which revealed the absence of any particles.

Micro-BCA assay (Thermo Scientific, Waltham, MA, USA) was performed as per the manufacturer’s protocol to determine the free protein concentration of various EV fractions (F1–F8), and the EV purity was evaluated by calculating the ratio of particle count determined by NTA to the free protein concentration determined by micro-BCA.

#### 5.3.2. Transmission Electron Microscopy (TEM)

Negative staining [34] was performed for LMSC- and LM-sEV fractions with higher particle content (F3). In addition, unconditioned sEV fractions were included as a negative control. For fixation, the grid surface was rendered hydrophilic by UV irradiation for 10 min. Equal volumes of the sEV solution and 4% paraformaldehyde (PFA) were mixed. An amount of 5 μL of the resuspension was deposited onto Formvar-carbon coated EM grids, with 2–3 grids prepared per sample. After a 5 min incubation in a dry environment, the grids were placed membrane-side down on 20 μL drops of PBS on parafilm. The grids were then sequentially placed on 20 μL drops of 1% glutaraldehyde for 5 min, followed by 20 μL drops of MQ-H_2_O for 2 min, repeated seven times. The grids were then transferred to 30 μL drops of 2% uranyl oxalate solution (pH 7) for 5 min in the dark. Finally, the grids were transferred to 30 μL drops of methyl cellulose-uranyl acetate for 10 min on ice in the dark. Excess fluid was blotted using stainless steel loops and Whatman filter paper, and the grids were air-dried for 5–10 min before storage in grid storage boxes for eventual observation under an electron microscope at 80 kV.

#### 5.3.3. Western Blot Analysis

WB analysis was performed as described previously [85]. Equal amounts of EV fractions (15 µL (~10 µg); F1–F8) or whole-cell lysate (10 µg; prepared using RIPA buffer: ThermoFisher Scientific, Waltham, MA, USA) were solubilized using 4× Laemmli buffer (Biorad, Feldkirchen, Germany) in reducing conditions at 95 °C for 10 min. Samples were then separated by SDS-PAGE under reducing conditions, and immunoblot analyses were performed using antibodies (Appendix A) followed by horseradish peroxidase-labeled anti-mouse or rabbit IgG (Jackson ImmunoResearch). Protein bands were visualized using the enhanced chemiluminescence Western blot detection reagent (GE Healthcare) and the fusion fx Imager/fusion software (Fusion FX7 Edge 18.05, Vilber Lourmat; https://www.vilber.com/fusion-fx/).

### 5.4. EV Analysis with the Multiplex Bead-Based Platform

LMSC- and LM-sEVs (F3) were screened for 37 different antigens using a MACSplex human exosome kit (Miltenyi, Bergisch Gladbach, Germany) as per manufacturer instructions. Briefly, 120 μL EV samples were loaded onto 1.5 mL tubes. To each tube, 15 μL of MACSPlex Exosome Capture Beads were added and tubes were incubated at room temperature on orbital shaker overnight (14–16 h) at 450 rpm. Subsequently, the beads were washed in MACSplex buffer. EVs bound by capture beads were stained for 1 h at room temperature on an orbital shaker at 450 rpm with 15 μL of MACSPlex Exosome Detection Reagent cocktail comprising CD9/CD63/CD81-APC antibody conjugates. After staining, beads were washed and analyzed by flow cytometry (BD Accuri C6 Plus) and post-acquisition analysis was performed using FlowJo software (FlowJo 10.2, Tree Star Inc., Ashland, OR, USA). A blank control of MACSPlex buffer, incubated with beads and detection antibodies, was used to measure the background signal. The median fluorescence intensity (MFI) of each surface marker was normalized by the mean MFI for specific markers (CD9, CD63, and CD81). All analyses were based on the normalized MFI (nMFI) values.

### 5.5. Proteomics

sEV samples (F3) were resuspended in lysis buffer (5% SDS, 50 mM triethyl ammonium bicarbonate (TEAB, T7408, pH 7.5; Sigma, Taufkirchen, Germany). Afterwards samples were sonicated using a Bioruptor device (Diagenode, Liège, Belgium). Samples were centrifuged at 13,000× *g* for 8 min and the supernatant was used in the following steps. Proteins were reduced using 5 mM tris (2-carboxyethyl) phophine hydrochloride (TCEP) (Sigma; 75259) for 10 min at 95 °C and alkylated using 10 mM 2-iodoacetamide (Sigma; I1149) for 20 min at room temperature in the dark. The following steps were performed using S-Trap micro filters (Protifi, Huntington, NY, USA) according to the manufacturer’s protocol. Briefly, a final concentration of 1.2% phosphoric acid and then six volumes of binding buffer (90% methanol; 100 mM TEAB; pH 7.1) were added. After gentle mixing, the protein solution was loaded onto an S-Trap filter and spun at 2000 rpm for 0.5–1 min. The filter was washed three times using 150 μL of binding buffer. Sequencing-grade trypsin (1:25 enzyme: protein ratio; Promega, Mannheim, Germany) diluted in 20 µL digestion buffer (50 mM TEAB) was added to the filter and digested at 47 °C for 1 h. To elute peptides, three buffers were applied step-wise: (a) 40 μL 50 mM TEAB, (b) 40 µL 0.2% formic acid in H_2_O, and (c) 50% acetonitrile and 0.2% formic acid in H2O. The peptide solutions were combined and dried in a SpeedVac.

Peptides were analyzed with the Evosep One system (Evosep Biosystems, Odense, Denmark) coupled to a timsTOF fleX mass spectrometer (Bruker, Bremen, Germany). An amount of 500 ng of peptides was loaded onto Evotips C18 trap columns (Evosep Biosystems, Odense, Denmark) according to the manufacturer’s protocol. Peptides were separated on an EV1137 performance column (15 cm × 150 µm, 1.5 µm, Evosep) using the standard implemented 30 SPD method with a gradient length of 44 min (buffer A: 0.1% *v*/*v* formic acid, dissolved in H_2_O; buffer B: 0.1% *v*/*v* formic acid, dissolved in acetonitrile). The timsTOF fleX mass spectrometer was operated in the DDA-PASEF mode. MS and MS/MS spectra were acquired in an *m*/*z* range from 100 to 1700. Ion mobility resolution was set to 0.60–1.60 V·s/cm over a ramp time of 100 ms and an accumulation time of 100 ms. The data-dependent acquisition was performed using 10 PASEF MS/MS scans per cycle with a near 100% duty cycle. An active exclusion time of 0.4 min was applied to precursors that reached 20,000 intensity units. The collision energy was programmed as a function of ion mobility, following a straight line from 20 eV for 1/K0 of 0.6 to 59 eV for 1/K0 of 1.6. The TIMS elution voltage was linearly calibrated to obtain 1/K0 ratios using three ions from the ESI-L TuningMix (Agilent, Böblingen, Germany) (*m*/*z* 622, 922, 1222).

### 5.6. Bioinformatics

Raw data were analyzed with MaxQuant (v 2.1.4.0) and the built-in Andromeda peptide search engine [86]. The false discovery rate (FDR) at both the protein and peptide level was set to 1%. Two missed cleavage sites were allowed, no variable modifications were allowed, and carbamidomethylation of cysteines was set as fixed modification. For label-free quantification (LFQ), the MaxLFQ algorithm was applied using standard settings. Only unique peptides were used for quantification. The Human-EBI-reference database was downloaded from https://www.ebi.ac.uk/ accessed on 3 March 2022. LFQ intensity was used to quantify the proteins, and those lacking LFQ intensity were eliminated. Gene Ontology (GO) enrichment analysis was conducted using the clusterProfiler (4.8.3) package [87] in the R (4.3.1) statistical environment [88]. The enrichplot (1.20.3) [89], pathview (1.40.0) [90], ggplot2 (3.4.4) [91], ComplexHeatmaps (2.16.0) [92], and ggVennDiagram (1.2.3) packages [93] were used for data visualization.

### 5.7. EV Internalization Assay

sEV were fluorescently labeled with the lipophilic dye Memglow^TM^ 488, which is incorporated into lipid membranes. The sEV (~10 µg, F3) were incubated with 100 nM dye, protected from light at room temperature for 5 to 10 min. To remove the unbound dye, the labeled sEV were diluted with filtered PBS (make up to 1.5 mL) and concentrated to 100 µL using centrifugation with a proteus X-spinner 2.5 (100 kDa) filter at 2.500× *g* and 4 °C. Particles from unconditioned media were labeled as a negative control.

LEPC (P1) were seeded at a density of 100,000 cells/well in a 12-well plate. After 24 h, cells were incubated with fluorescently labeled sEV for 3 or 6 h. The cells were then washed with PBS twice to remove non-internalized sEV and trypsinized using 0.05% Trypsin-EDTA. Data were analyzed by flow cytometry FACSCantoII (BD Biosciences, Heidelberg, Germany) by using FACS Diva software (BD Pharmingen; BD Bioscienes, Heidelberg, Germany). Before analysis, cells were resuspended in PBS with DAPI to exclude dead cells from quantification. Post-acquisition analysis was performed using FlowJo software (Tree star, Inc., Ashland, OR, USA).

Additionally, LEPC were seeded at a density of 40,000 cells/well on 4-well chamber slides (Lab-Tek, Thermo Scientific, Dreieich, Germany). After 24 h, cells were incubated with fluorescently labeled sEV for 6 h. The cells were then washed with PBS twice to remove non-internalized sEV and fixed with 4% paraformaldehyde/PBS solution for 15 min at room temperature. Post-fixation, the cells were washed with PBS and stained with Phalloidin-TRITC for 30 min. Slides were washed with PBS and mounted using Vectashield antifade mounting medium with DAPI (Vector, Burlingame, CA, USA). Confocal fluorescence microscopy (TCS SP-8, Leica, Wetzlar, Germany) was used to image the fluorophore signals.

### 5.8. Immunohistochemistry

The immunostaining of frozen sections was carried out following the protocol detailed in our previous publication [94].

### 5.9. Statistics

The experiments were conducted with a minimum of four independent experiments. The data, presented as mean ± standard deviation (SD), were statistically analyzed using the GraphPad Instat software package for windows (Version 6.0; Graphpad Software Inc., La Jolla, CA, USA). The statistical significance was evaluated using Wilcoxon signed-rank test or Mann–Whitney U test as appropriate, with *p* value < 0.05 considered statistically significant.

## Figures and Tables

**Figure 1 cells-13-00623-f001:**
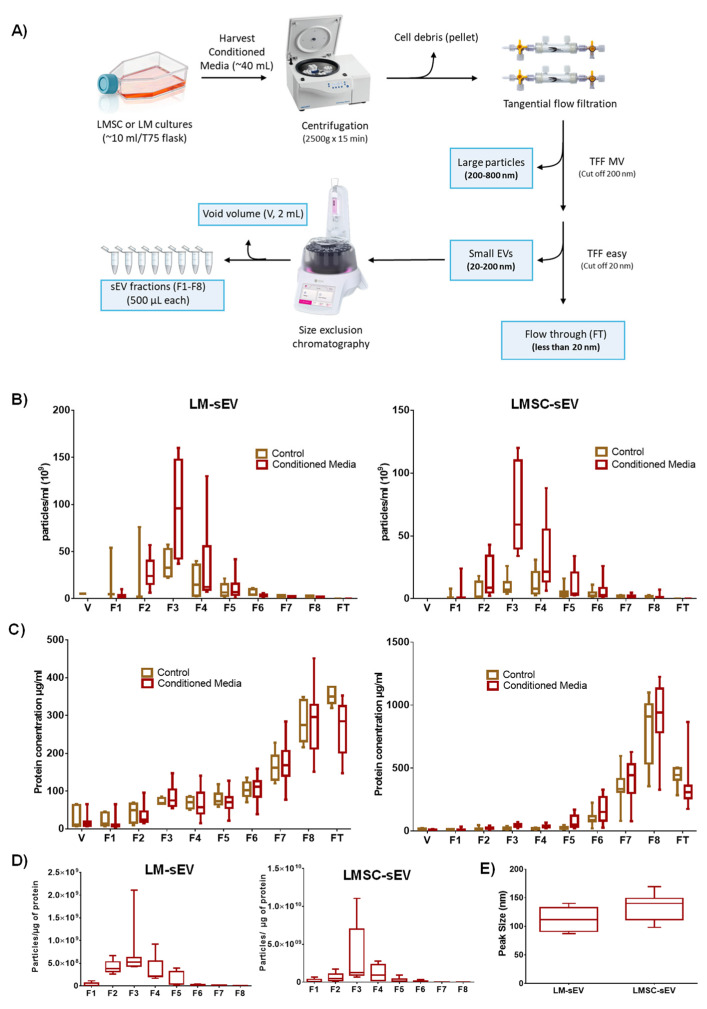
**Enrichment and quantification of small extracellular vesicles (sEV):** (**A**) Workflow used for enrichment of sEVs from conditioned media (~40 mL) of limbal mesenchymal stromal cells (LMSC) and limbal melanocytes (LM). (**B**) Nanoparticle tracking analysis showing particle counts in LM-sEV and LMSC-sEV fractions (F1–F8), void volume (V), and flow-through (FT) of conditioned and unconditioned media. Data presented as min-to-max whisker box plots (n = 10/cell type). (**C**) Protein concentration analysis by micro-BCA assay showing higher protein levels in later fractions (F7, F8) and flow-through (FT) compared to earlier fractions for both LM-sEVs and LMSC-sEVs (n = 10/cell type sEVs). Data presented as min-to-max whisker box plots (n = 10/cell type). (**D**) Particle-to-protein ratio analysis indicating highest purity in fraction 3 (F3) for both LM-sEVs and LMSC-sEVs (n = 10/cell type). Data presented as min-to-max whisker box plots (n = 10/cell type). (**E**) Size distribution of LM-sEV and LMSC-sEV particles in fraction 3 (F3) measured by nanoparticle tracking analysis. Data presented as min-to-max whisker box plots (n = 10/cell type).

**Figure 2 cells-13-00623-f002:**
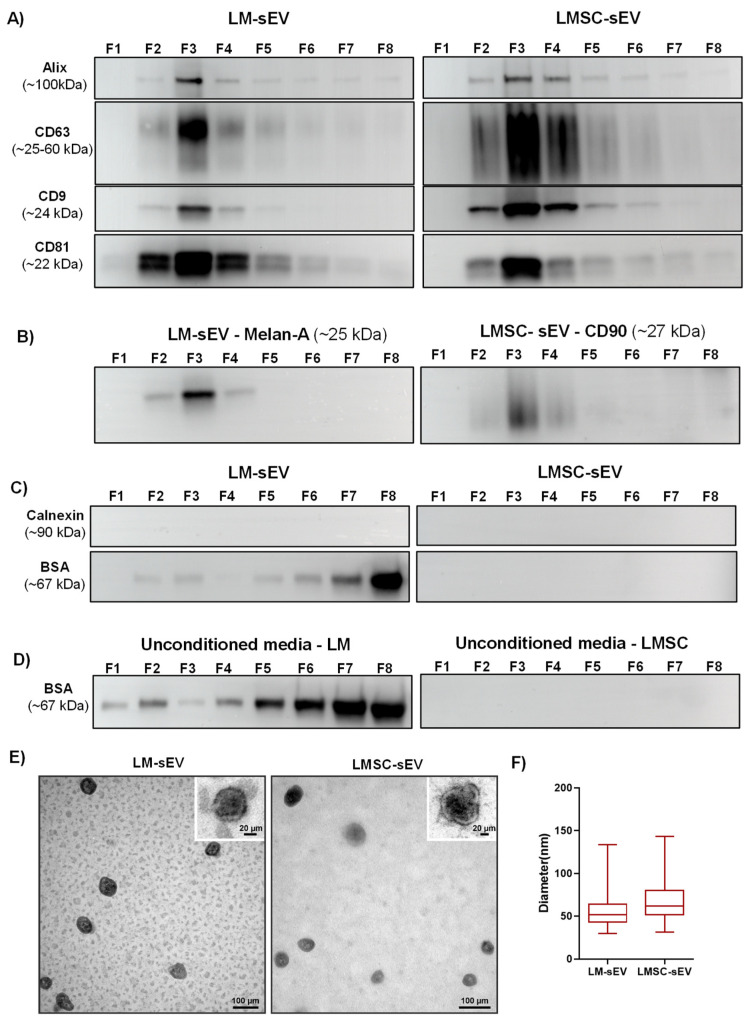
**Characterization of small extracellular vesicles (sEV):** (**A**) Western blot analysis of EV-specific markers in fractions F1–F8 from LM- and LMSC-sEVs. Enrichment of EV markers was predominantly observed in F3. Uncropped versions of the Western blot are shown in Appendix A. (**B**) Cell-specific markers CD90 (LMSC) and Melan-A (LM) were detected in F2–F4, with highest levels in F3. Uncropped versions of the Western blot are shown in Appendix A. (**C**) Absence of the endoplasmic reticulum marker calnexin (CANX) across all fractions indicates lack of cellular contamination. Residual bovine serum albumin (BSA) was detected in LM-sEV fractions. Uncropped versions of the Western blot are shown in Appendix A. (**D**) BSA was present in unconditioned media controls for LM-sEVs but not LMSC-sEVs. Uncropped versions of the Western blot are shown in Appendix A. (**E**) Transmission electron microscopy images of F3 fractions showing classical EV morphology with intact membranes. (**F**) Size distribution box plots of LM-sEVs and LMSC-sEVs in F3 based on TEM analysis. Data presented as min-to-max whisker box plots (n = 3/cell type).

**Figure 3 cells-13-00623-f003:**
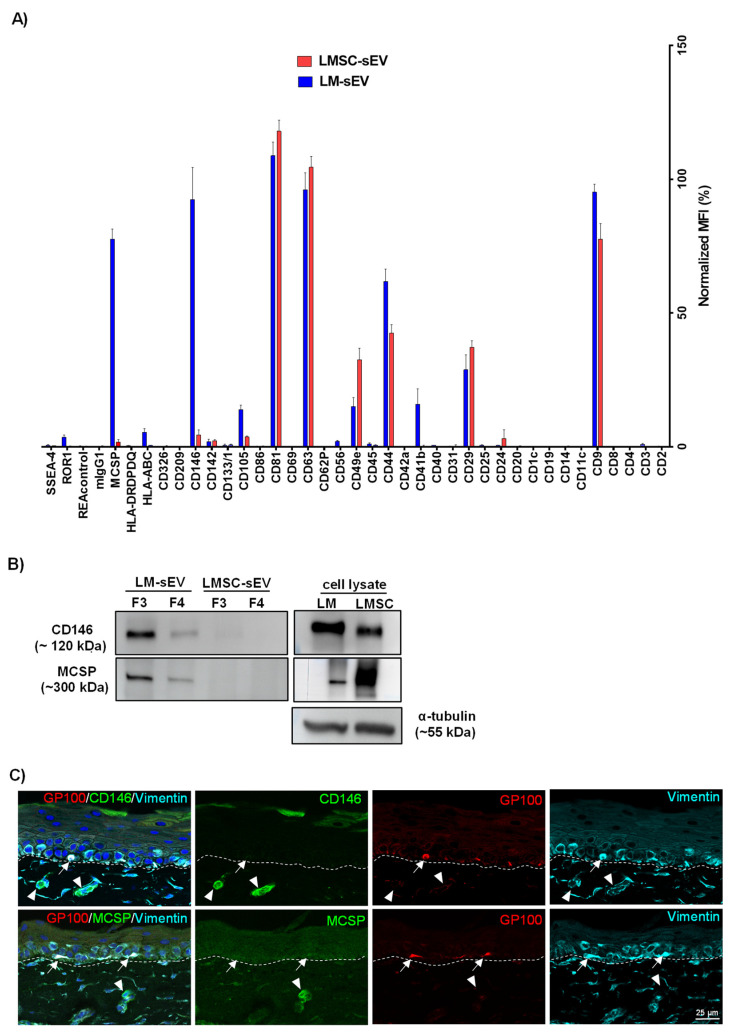
**Multiplex bead-based flow cytometry assay:** (**A**) Column graphs showing normalized median fluorescence intensity (nMFI) values for various EV surface antigens detected by multiplex bead array analysis of fraction 3 (F3) from limbal melanocyte-derived sEVs (LM-sEVs) and limbal mesenchymal stromal cell-derived sEVs (LMSC-sEVs). Data presented as mean ± standard deviation (SD), n = 5/cell type. (**B**) Western blot validation confirming predominant expression of CD146 and MCSP in LM-sEVs (F3) compared to LMSC-sEVs. Cell lysate of both LMSC and LM showing the expression of CD146 and MCSP. Uncropped versions of the Western blot are shown in Appendix A. (**C**) Immunohistochemical staining of organ-cultured limbal tissue sections showing absence of CD146 (green) and MSCP (green) in melanocytes (arrows, red, gp100^+^) and stromal cells (cyan, vimentin^+^). Vessels in the stroma stained for CD146 (arrow heads, green) and MCSP (arrow heads, green). Dashed line represent the basement membrane. Nuclei are counterstained with DAPI (blue).

**Figure 4 cells-13-00623-f004:**
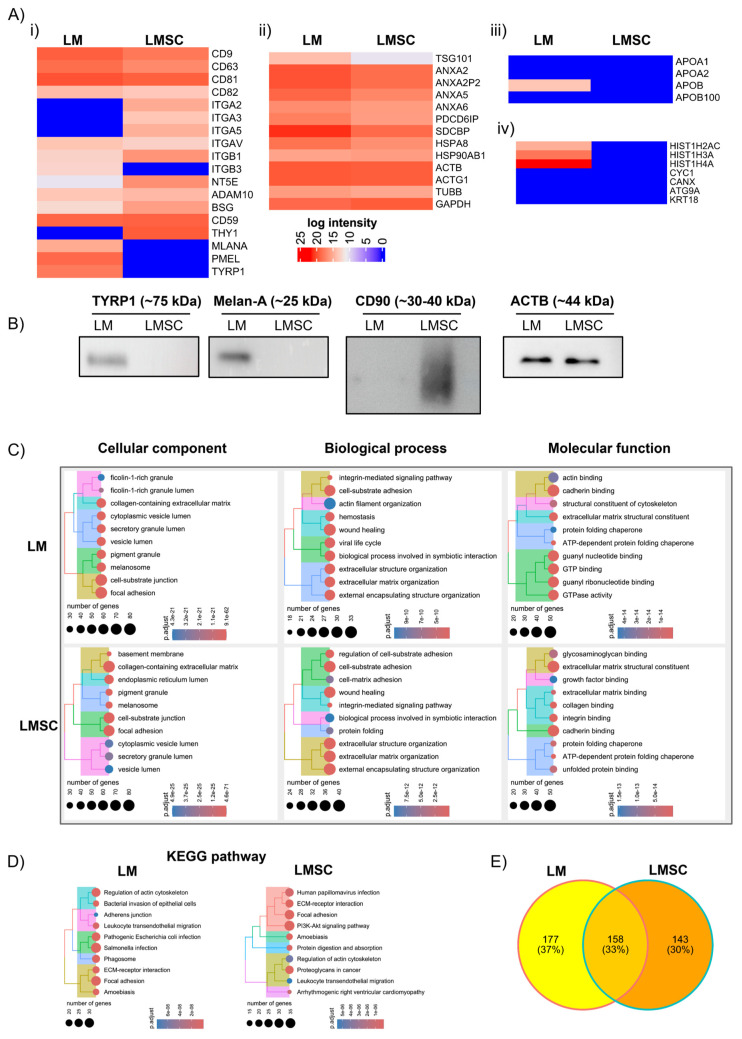
**Proteomic characterization of small extracellular vesicles (sEV):** (**A**) Heatmaps showing protein intensity profiles (based on iBAQ mean intensity, n = 5) of (**i**) plasma membrane and endosomal proteins, (**ii**) cytosolic proteins, (**iii**) apolipoproteins, and (**iv**) cellular contaminants across LMSC-sEV and LM-sEV samples. We employed relative quantification (iBAQ) to compare the abundance levels of known EV markers and potential contaminants (such as serum components or cytosolic proteins) within the sample. (**B**) Western blot validation confirming the presence of CD90 (LMSC), Melan-A and TYRP1 (LM), and β-actin in fraction 3 (F3) of respective sEV populations. Uncropped versions of the Western blot are shown in Appendix A. (**C**,**D**) Top 10 enriched Gene Ontology (GO) terms for (**C**) Biological Process, Molecular Function, Cellular Component, and (**D**) KEGG pathways in LM-sEV and LMSC-sEVs. (**E**) Venn diagram showing the number of unique and shared proteins identified in LM-sEVs and LMSC-sEV.

**Figure 5 cells-13-00623-f005:**
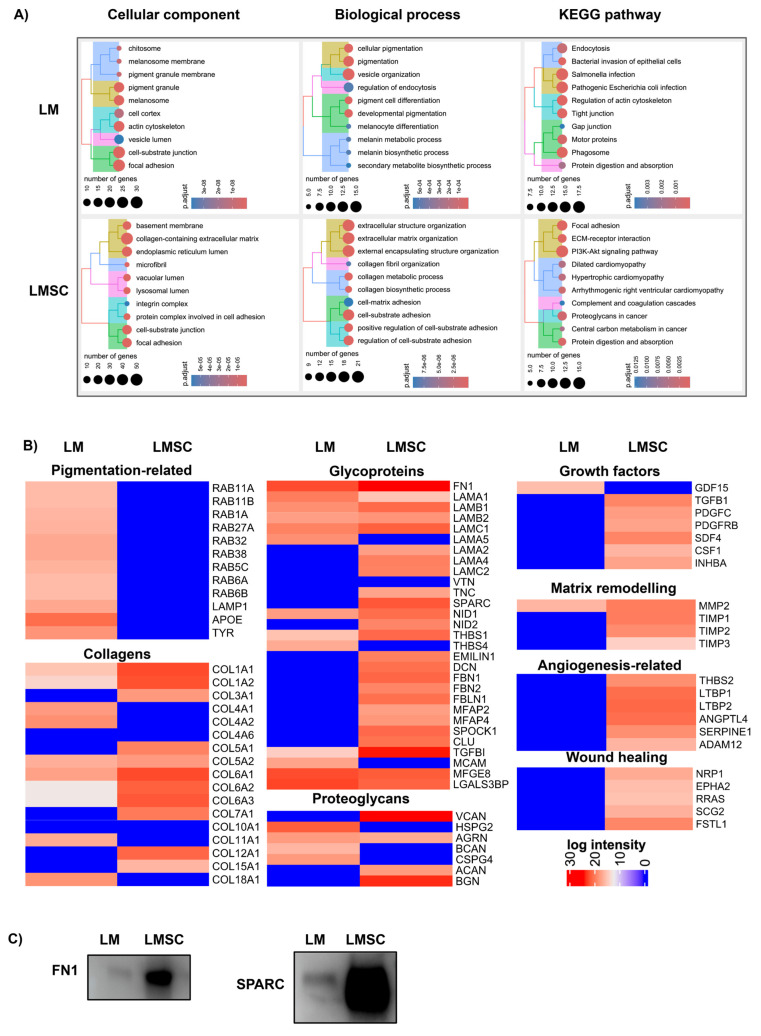
**Distinct protein signatures in limbal melanocyte (LM)-derived small extracellular vesicles (sEV) and limbal mesenchymal stromal cell (LMSC)-derived small extracellular vesicles (sEV)**. (**A**) Gene ontology analysis highlighting the unique biological processes, cellular components, molecular functions, and KEGG pathways enriched in proteins found exclusively in LM-sEVs versus LMSC-sEVs. (**B**) Heatmaps showing selected proteins (based on LFQ mean intensity, n = 5) uniquely present in LM-sEVs versus LMSC-sEVs, categorized based on their roles in melanogenesis, extracellular matrix composition and remodeling, angiogenesis, and cell migration. (**C**) Western blot validation confirming the presence of fibronectin (FN1) and SPARC in LMSC sEV populations. Uncropped versions of the Western blot are shown in Appendix A.

**Figure 6 cells-13-00623-f006:**
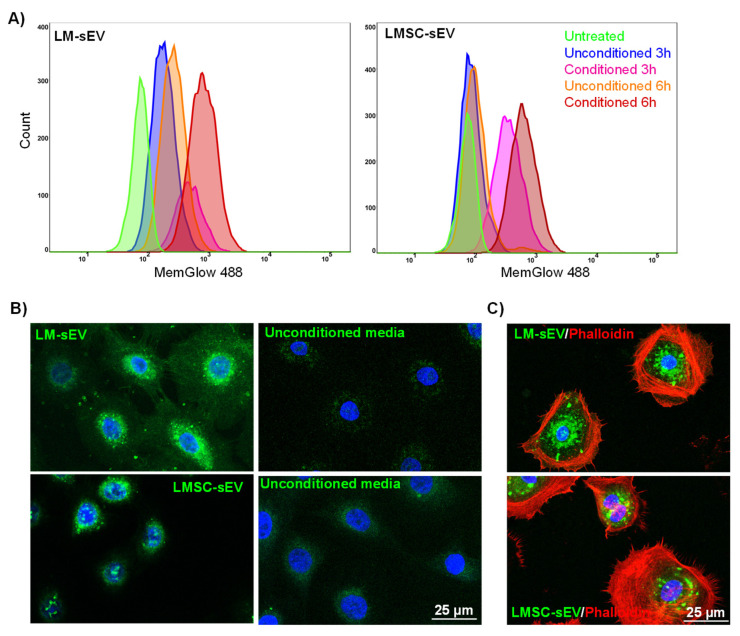
**Uptake of isolated small extracellular vesicles (sEVs) by recipient cells:** (**A**) Flow cytometry analysis demonstrating increased fluorescence intensity in limbal epithelial progenitor cells (LEPC) treated with limbal mesenchymal stromal cell derived small extracellular vesicles (LMSC-sEVs) or limbal melanocyte derived small extracellular vesicles (LM-sEVs) for 3 or 6 h compared to untreated LEPC or LEPC treated with sEVs derived from unconditioned media. (**B**) Confocal microscopy images showing the localization of fluorescently labeled LMSC-sEVs and LM-sEVs (green) in the perinuclear region of LEPC after 6 h of incubation. Nuclei are counterstained with DAPI (blue). (**C**) LEPC co-stained with Phalloidin-TRITC (red) to visualize F-actin. Nuclei are counterstained with DAPI (blue).

## Data Availability

The datasets generated during and/or analyzed during the current study are available from the corresponding author on reasonable request.

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
