# Peer review of "Enrichment, Characterization, and Proteomic Profiling of Small Extracellular Vesicles Derived from Human Limbal Mesenchymal Stromal Cells and Melanocytes"

_cells, 2024, doi:10.3390/cells13070623_

Round 1
Reviewer 1 Report
Comments and Suggestions for Authors
This study provides fundamental insights into the LM- and LMSC-derived sEV populations by particle and protein analysis. The experiments and results is detailed, however, the innovativeness of this study still needs to be improved. Comments could be found as follow:
1)P1 cells were used for this study, thus, the purity of these cell types (especially the LMSC) is the key for the downstream analysis.
2) The distinct functions of these two sEV populations are associated to their cell of origin. Please provide more discussions about the relationship between sEVs and their cell type.
3) Did sEVs have any functional or state effects on recipient cells?
4) Comparative analysis between this study and previously-reported LMSC study is also encouraged.
Reviewer 2 Report
Comments and Suggestions for Authors
The research article by Kistenmacher et al investigate EVs in the limbus through a detailed characterization. The findings of such a research might help to highlight the role of EVs in cell-cell communication within different cellular components of the same organs and on how these orchestrate different cellular processes, such as differentiation and tissue regeneration.
Nevertheless the article is sometime confused, hampering the possibility for the reader to appreciate the results of the research.
First, although authors decided to merge results and discussion, I would suggest to split the two in order to make the manuscript more understandable.
Moreover, in Figure 1 the letters in the image and in caption are not the same. It would be interesting to know if the mean dimensions of EVs in the other fractions were similar to those in fraction 3, it can be reported, at least in the supplementary material.
Figure 3, I would suggest to add the house keeping in the western blot picture, in A authors show the expression of CD41b in LM EVs, CD41b is considered a unique marker of platelets, please discuss this result further.
Figures 4 and 5 are quite confused, letters are not correct, in some graphs there is a title, for instance ‘’molecular function’’, for other there is not, for instance ‘’KEGG pathway’’. Moreover, it is not clear that in figure 4 authors analyse a relative quantification, while in figure 5 they do a quantification. Beside these qualitative lacks I think that data analysis should be improved. I suggest to perform a PCA analysis for both qualitative and quantitative proteome analysis and for the multiplex flow cytometry analysis in figure 3, add also to the PCA analysis which are the main distinctive protein markers for each component, in order to visualise how the groups segregates and which are the distinctive proteins. Moreover, since authors state to find a protein signature, I would suggest to do an unsupervised clustering and show the produced heatmaps. Multiplex flow cytometry analysis data should be correlated with MS data in order to identify which distinctive elements were enriched using both approaches. Finally also pathway and GO analysis are not easily readable as they are presented, I would suggest to mix together the enriched elements of both LM and LMSC, for example in the same heatmap or dot plot, using the first ten outputs for both categories. In this way, it is possible to easily compare which are the differences between the two groups and for example which pathways are co-enriched.
In the discussion I would suggest some hints on how LM EVs might protect the cornea from UV induced damage, focusing on the effect that UV light was proved to have in the processes of cell death, proliferation, differentiation and eventually pterygium/ocular surface cancer development, please refer to https://www.sciencedirect.com/science/article/pii/S0014483518305025 and https://www.mdpi.com/2079-7737/12/2/265 as example of UV response in the cornea.
Comments on the Quality of English LanguageSome typos throughout the text. Please check it before resubmitting.
Round 2
Reviewer 1 Report
Comments and Suggestions for Authors
Comments have been responded well in the revised manuscript.
Reviewer 2 Report
Comments and Suggestions for Authors
Consider to move some of the pictures regarding the heat maps and PCA from the supplementary to the main text.